# Isolation and Antibacterial Activity of Indole Alkaloids from *Pseudomonas aeruginosa* UWI-1

**DOI:** 10.3390/molecules25163744

**Published:** 2020-08-17

**Authors:** Antonio Ramkissoon, Mohindra Seepersaud, Anderson Maxwell, Jayaraj Jayaraman, Adesh Ramsubhag

**Affiliations:** 1Department of Life Sciences, The University of the West Indies, St. Augustine, Trinidad and Tobago; antonioramkissoon01@gmail.com (A.R.); Jayaraj.Jayaraman@sta.uwi.edu (J.J.); 2Moderna Inc., 200 Tech Square, Cambridge, MA 02139, USA; m.seepersaud101@gmail.com; 3Department of Chemistry, The University of the West Indies, St. Augustine, Trinidad and Tobago; andersonmaxw@gmail.com

**Keywords:** *Pseudomonas aeruginosa*, antibacterial compounds, indole alkaloids, natural products, tris(1*H*-indol-3-yl) methylium, bis(indol-3-yl) phenylmethane, indolo (2, 1b) quinazoline-6, 12 dione

## Abstract

In this study, we report the first isolation of three antibiotic indole alkaloid compounds from a Pseudomonad bacterium, *Pseudomonas aeruginosa* UWI-1. The bacterium was batch fermented in a modified Luria Broth medium and compounds were solvent extracted and isolated by bioassay-guided fractionation. The three compounds were identified as (**1**) tris(1H-indol-3-yl) methylium, (**2**) bis(indol-3-yl) phenylmethane, and (**3**) indolo (2, 1b) quinazoline-6, 12 dione. A combination of 1D and 2D NMR, high-resolution mass spectrometry data and comparison from related data from the literature was used to determine the chemical structures of the compounds. Compounds **1–3** were evaluated in vitro for their antimicrobial activities against a wide range of microorganisms using the broth microdilution technique. Compounds **1** and **2** displayed antibacterial activity against only Gram-positive pathogens, although **1** had significantly lower minimum inhibitory concentration (MIC) values than **2**. Compound **3** displayed potent broad-spectrum antimicrobial activity against a range of Gram positive and negative bacteria. Several genes identified from the genome of *P. aeruginosa* UWI-1 were postulated to contribute to the biosynthesis of these compounds and we attempted to outline a possible route for bacterial synthesis. This study demonstrated the extended metabolic capability of *Pseudomonas aeruginosa* in synthesizing new chemotypes of bioactive compounds.

## 1. Introduction

The discovery of antibiotics in the early 20th century led to a revolution in modern medicine by providing cures for many prevalent diseases that were not treatable at that time. Resistance to these drugs, however, followed soon thereafter and has limited the useful lifespan of therapeutic antibiotic agents. It is therefore essential for novel antimicrobials to be developed for treatment with the ever-increasing threat of antimicrobial resistance [1].

Historically, natural products from microorganisms and plants were the mainstay for the discovery of antibiotic drugs [2]. These molecules possess intrinsic bioactivities that evolution has shaped to improve the fitness of the host organisms in its environment [3]. In the natural setting, antimicrobial compounds protect the host from organisms competing for nutrients and space [4]. The discovery pipeline from these traditional sources has yielded little returns within recent times and the search for novel chemotherapeutics is now increasingly being focused on new techniques such as combinatorial chemistry, rational drug design, and genome mining. However, there is still potential for discovering novel antibiotic drug compounds from traditional natural product isolation and characterization [5]. This holds true especially for bacteria, whose genomes are considered plastic and variations even on the strain level can result in altered secondary metabolite profiles [6]. Secondary metabolism is not considered an essential process for microbial growth, but it has been well established that it contributes to an organism’s fitness. Whilst primary metabolism is strictly conserved within the genome, secondary metabolism is widely divergent and is often strain specific [7]. This type of metabolism is highly influenced by microbial biogeography, and hence a strain that is found in one location may not produce the same metabolites as the same organism found in another location [8].

Pseudomonads have been one of the most bioactive bacterial families, with over 600 bioactive compounds being reported. They are also the most commonly isolated antibiotic producers from natural environments, as they can grow in many substrates and are very easy to culture. Amongst the pseudomonads, *Pseudomonas aeruginosa* is regarded as a very prolific antibacterial species, known to produce a wide variety of compounds with inhibitory properties. These include the phenazines, quinolones, rhamnolipids, hydrogen cyanide, lectins, pyrroles, pseudanes, indoles, and various amino acids and peptides [9].

Two important factors influencing antibiotic production in *P. aeruginosa* include the genetic capacity of strains to create specific bioactive secondary metabolites and fermentation conditions that enable antibiotic production. This is because the biosynthesis of small molecules is often coded by biosynthetic gene clusters and the genes are often cryptic under standard laboratory conditions but can be turned on by various cues, which are usually specific signal molecules [10,11].

In an unpublished report, we demonstrated that *Pseudomonas aeruginosa* UWI-1 had the genetic capacity to create numerous bioactive compounds as examination of its genome revealed that the bacterium had a large proportion of genes dedicated to secondary metabolite production. As part of our investigation of species indigenous to Trinidad and Tobago, *P. aeruginosa* UWI-1 was evaluated for antibiotic production. Medicinal chemistry efforts on this strain has resulted in the isolation of three unique indole based antimicrobial compounds through bioassay guided fractionation of fermented broth cultures. Also, through gene sequencing and annotation coupled with determination of gene functions from other reports, the probable biosynthetic mechanisms for the production of these compounds were established. This study represents the first report on isolation of these compounds from any Pseudomonad.

## 2. Results

### 2.1. Identification of Antibacterial Compounds

*P. aeruginosa* UWI-1 was cultured in modified Luria broth (mLB) and the fermentation products of the bacteria were isolated and screened for antimicrobial activity using a bioassay-guided isolation protocol. The three antibacterial compounds isolated from *P. aeruginosa* UWI-1 are shown in Figure 1.

Based on preliminary ^1^H and ^13^C NMR investigations, compounds **1** and **2** appeared to be structurally related as both contained key proton and carbon shift patterns indicative of the indole ring (Table 1). Compound **1** was originally isolated as brown crystals and the melting point was recorded as 240–242 °C. Electrospray Ionization (ESI) mass spectrum showed a pseudomolecular ion at *m*/*z* 360.439 calculated for C_25_H_18_N_3_. One broad singlet at δ_H_ 8.10 (br s, 3× NH), two doublets at 7.65 (3H, d, *J* = 8 Hz) and 7.39 (3H, d, *J* = 8 Hz), and two triplets at δ_H_ 7.19 (3H, t, *J* = 8 Hz) and 7.11 (3H, t, *J* = 8 Hz) in the ^1^H NMR spectrum revealed that the compound contained the indole ring. The δ_C_ 109.5 with no attached protons revealed the presence of a methylium cation to which the carbon 3 of the indole rings were attached. Based on the coupling patterns and the molecular mass, we concluded that the compound was composed of an indole trimer and the data matched that of Turbomycin A [12].

Compound **2** was isolated as an amorphous red solid with a molecular formula of C_23_H_18_N_2_ determined by HR-ESIMS showing a pseudomolecular ion at *m*/*z* 321.1363 [M + H] ^+^. Melting point was measured at 125–126 °C. The analysis of the ^1^H and COSY spectrum of **2** (Table 1) revealed the presence of two three-substituted indole rings, with proton signals at δ_H_ 7.92 (2H, brs, NH), two doublets at 7.35 (2H, *J* = 7.5 Hz) and 7.40 (2H, *J* = 7.5 Hz), and two triplets at δ_H_ 7.14 (2H, *J* = 7.5 Hz) and 7.00 (2H, *J* = 7.5 Hz). In addition, the proton signals at δ_H_ 7.75 (2H, *J* = 7.5 Hz), δ_H_ 7.24 (2H, *J* = 7.5 Hz), and δ_H_ 7.13 (2H, *J* = 7.5 Hz) are reminiscent of a phenyl ring. The HMBC and HSQC correlation allowed the connection of these moieties to a methyl carbon at δ_C_ 40.2; δ_H_ 5.89 (1H, s). Both ^1^H/^13^C NMR data compared well with literature data [13], while ESI mass spectrum was consistent with the molecular formula of bis(indol-3-yl)phenylmethane.

Compound **3** (tryptanthrin) was isolated as pale-yellow needle crystals. This compound was also identified using 1D and 2D NMR data (Table 2) and the ESI mass spectrum which showed a pseudomolecular ion at *m*/*z* 271.23 [M + Na]^+^ consistent with its molecular formula, C_15_H_10_N_2_O_2_. The melting point of this compound was 266–267 °C. The analysis of the ^1^H and COSY spectrum of **3** revealed the presence of two *ortho*-disubstituted benzene rings by assignment of two spin systems. The first ring system was at **H-1–H-2–H-3–H-4** [δ_H_ 8.42 (1H, *J* = 7.9 Hz, *d*), δ_H_ 7.66 (1H, *J* = 7.9 Hz, t), δ_H_ 7.84 (1H, *J* = 7.9 Hz, t), δ_H_ 8.02 (1H, *J* = 7.9 Hz, d)] and the second was at **H-7–H-8–H-9–H-10** [δ_H_ 7.90 (1H, *J* = 8.1 Hz, d), δ_H_ 7.42 (1H, *J* = 8.1 Hz, t), δ_H_ 7.78 (1H, *J* = 8.1 Hz, t), δ_H_ 8.62 (1H, *J* = 8.1 Hz, d)]. Analysis of the HMBC and HSQC data confirmed the presence of the two benzene systems, as well as a characteristic amido carbon (C-12) at δ_C_ 158.1, an imino (C-5a) group at δ_C_ 144.1 and a carbonyl (C-6) group δ_C_ 182.5.

### 2.2. Antibacterial Activity of Isolated Compounds

The antibacterial activity of the three compounds isolated from *P. aeruginosa* UWI-1 is shown in Table 3. Compound **1** (tris(1H-indol-3-yl) methylium) displayed broad-spectrum antimicrobial activity. The compound was able to inhibit the growth of Gram-positive bacteria at concentrations (minimum inhibitory concentration (MIC) = 1–16 µg mL^−1^) comparable to the antibiotic control, Kanamycin. It also performed better against *C. diphtheria* and *S. pyogenes* than Erythromycin. Compound **1** inhibited the Gram-negative bacteria as well, but only at higher concentrations (MIC = 32–128 µg mL^−1^). When compared to the antibiotic controls, it was observed to be more potent when compared to Erythromycin, but less so compared to Kanamycin. In most cases the minimum bactericidal concentration (MBC) ≤ 4X MIC, which meant that this compound exerts a bactericidal mechanism of action.

Compound **2** (bis(indol-3-yl)phenylmethane) displayed poor antimicrobial activity against the panel of pathogens used. Inhibition was limited to Gram-positive microbes, and the MICs were comparatively higher (32–256 µg mL^−1^) than other test compounds as well as the antibiotic controls. There was no inhibition by the concentrations of **2** (bis(indol-3-yl)phenylmethane) tested against Gram-negative bacteria. Even though the MICs were relatively higher than those of clinical relevance, the MBC concentrations were at most double that of the respective MIC, which could mean that Compound **2** also displays a bactericidal mechanism of action.

Assessment of compound **3** (tryptanthrin’s) antimicrobial activity showed broad-spectrum inhibition of bacterial pathogens, though it was more potent against the Gram-positive bacteria than the Gram-negatives. The MIC values against Gram positive bacteria ranged between 1–16 µg mL^−1^, whereas MIC values against the Gram-negatives ranged between 2–32 ug mL^−1^.

### 2.3. Biosynthesis of Antibacterial Compounds

To propose a biosynthetic route for the production of the three compounds isolated, the genome of *Pseudomonas aeruginosa* UWI-1 was examined to identify the putative genes involved in key biosynthetic steps (Figure 2 and Figure 3).

Compounds **1** (Turbomycin A) and **2** (bis(indol-3-yl) phenylmethane) are both triaryl indole alkaloids. The sub-cloning experiments that led to the first reported isolation of turbomycins revealed that the microbial production of these triaryl indole based compounds was due of catalytic activity of 4-hydroxyphenylpyruvate dioxygenase (4HPPD) [12]. However, since that time, the chemical route for the synthesis of these compounds has been developed and it is now known that Turbomycin A (**1**) is formed from the condensation reaction of two molecules of indole and one indole-3-carboxaldehyde, followed by oxidation of the methyl carbon to a methylium moiety. Similarly, the bis(indolylmethanes) (**2**) are formed by condensation of two indoles with a benzaldehyde group [14,15].

Genomic analysis of *P. aeruginosa* UWI-1 shows that the bacterium contains the 4HPPD gene on the antisense strand of the genome between 3,233,515 and 3,234,588 bp, which may have the propensity to generate homogentisate (HGA)-melanin and catalyze the formation of both **1** and **2**. However, there were no putative hits for the tryptophanase enzyme that is capable of breaking down tryptophan into indole but it is likely that indole was obtained during tryptophan synthesis.

The bacterium possesses the biosynthetic means of synthesizing indole-3-carboxyaldehyde, as this compound is an intermediate in the synthesis of indole-3-acetic acid by pseudomonads. This was necessary for the production of compound **1**. The conversion is facilitated by tryptophan-2-dioxygenase found on the + strand between 1,269,361–1,270,227 bp.

The benzaldehyde required for the production of **2** may have been as a result of the metabolic products from polycyclic aromatic hydrocarbon degradation. The naphthalene dioxygenase (NDO) enzyme from *Pseudomonas* sp. has been shown to oxidize toluene into benzyl alcohol and benzaldehyde by Lee and Gibson [16]. NDO is present in the UWI-1 genome on the negative strand between 3,527,121–3,528,047 bp.

Compound **3** (tryptanthrin) may have been formed from the reaction of anthranilic acid with isatin which cyclizes to form the quinazoline system [17]. It was found that *P. aeruginosa* UWI-1 contains the necessary genes for the production of both anthranilic acid and isatin. Anthranilic acid is formed from the conversion of chorismate by the enzyme anthranilate synthase (AS) during the synthesis of the *Pseudomonas* quinolone signal (PQS) [18]. Anthranilate synthase was found within the UWI-1 genome on the positive strand between 5,740,744−5,742,222 bp. For isatin production, a study reported by Takeshige et al. [15] showed that the enzyme inosine 5-monophosphate dehydrogenase (IMPD) was needed as it converts 3-oxyindole into isatin. Isatin has been also shown to be formed as a product of NDO activity on indole by O’Connor and Hartmans [19]. Both the IMPD gene (positive strand, 6,341,685–6,343,154 bp) and NDO gene (negative strand, 3,527,121–3,528,047 bp) were found within the UWI-1 genome.

## 3. Discussion

Pseudomonads have a historic precedence in the production of bioactive secondary metabolites. To date over 600 bioactive compounds have been isolated and identified from this genera, and further investigations will inevitably see the increase in this number. *Pseudomonas aeruginosa*, in particular, is known to produce several compounds that possess antibacterial activity.

This study reports the first production of tris(1*H*-indol-3-yl)methylium (turbomycin A), bis(indol-3-yl)phenylmethane, and indolo(2, 1b) quinazoline-6, 12 dione (tryptanthrin) by any Pseudomonad, one of the most widely occurring groups of bacteria worldwide. These compounds add to the plethora of antibacterial compounds known to be produced by *Pseudomonas* spp. and highlights the enhanced secondary metabolic capacity of this organism to create bioactive compounds which confers an ecological advantage.

Turbomycin A was previously reported as an antibacterial secondary metabolite from various microbial sources such as *Vibrio parahaemolyticus* and *Saccharomyces cerevisiae* [20]. Turbomycin A was first obtained in a soil metagenomics study by Gillespie et al. [12]. In that study, sequence analysis of BAC clones revealed that a single open reading frame was adequate to produce two pigmented species i.e., Turbomycin A and Turbomycin B from *E. coli*.

Bis(indolylmethanes) (BIMs) are a group of indole alkaloids which possess a basic skeleton of two indole groups bonded to a single carbon via the 3 and 3′ positions. Based on the substituents attached to the bridging methyl carbon atom, they are called by different names. These compounds are typically found in both marine and terrestrial organisms. Due to their wide applications in medicinal chemistry, drug discovery, and agrochemicals, the synthesis and isolation of BIMs have attracted the attention of several chemists over the last few years [21].

Tryptanthrin (indolo(2, 1b)quinazoline-6, 12 dione), is a naturally occurring indoloquinazoline alkaloid first discovered in the Chinese indigo plant *Isatis tinctoria* [22]. It has also been isolated in several different plant species including *Calanthe, Wrightia, Couroupita and Strobilanthus* sp. [23], and from the yeast *Candida lipolytica* when grown in medium containing excess tryptophan [24]. Trypanthrin and its analogues have attracted interest as potential therapeutic agents over the years due to their diverse and potent biological activities, coupled with their ease of synthesis [25]. Tryptanthrin has been reported to possess potent antibacterial activity against *Bacillus* and *Mycobacterium* sp., antifungal activity against *Trychophyton*, *Microsporum* and *Epidermophron* sp., and inhibit proliferation of *Leishmania donovani*, *Trypanosoma brucei*, and *Plasmodium falciparum*. It and its derivatives also display significant antagonistic activity against several human cancer cell lines [26].

Interestingly, all three bioactive compounds identified in this study are all indole alkaloids. Indole forms the core chemical component of a number of biologically active natural product molecules [27]. The indole nucleus is considered a privileged structure as it is an important ring system often associated with pharmaceutical development. The inherent biological activity possessed by the compounds identified in this study and many other indole alkaloid compounds is primarily through the number of chemical interactions that the indole core of these ligands are able to interact with target proteins. The presence of nitrogen atom in indole ring maintains the aromatic system and makes binding N-H acidic rather than nitrogen basic. The indole ring is able to form hydrogen bonds through the N-H moiety and π–π stacking or cation–π interactions, via the aromatic moiety [28].

Compounds **1**, **2**, and **3** all appear to have greater antibacterial potency against the Gram-positive bacteria than against the Gram-negative bacteria due to the lower minimum inhibitory values recorded. It is well established by many researchers that while the specific mechanisms of action of indole derivatives remain elusive, these compounds have good affinity to protein kinase enzymes. Zorgahi et al. [29] demonstrated the selective inhibition of Methicillin-resistant *Staphylococcus aureus* (MRSA) pyruvate kinase of various indole alkaloids. In another study, Eusynstyelamides A and B possessed inhibitory activity toward pyruvate phosphate dikinase of various bacterial species [30]. It has been reported that the indole pharmacore and its many derivatives are able to inhibit autophosphorylation events of protein kinases in various species [31,32,33,34].

It is with this in mind that we believe that the greater inhibition against the Gram-positive bacteria may be primarily due to the bacterial penetration. Gram-negatives pose a far greater threat than Gram-positive bacteria and they occupy nine of the twelve positions on the World Health Organization’s list of priority organisms for which new antimicrobials are needed [35]. Gram-negative bacteria are harder to treat with antibiotics as their outer membrane serves as a permeability barrier, and hence, can exclude certain antibiotics from penetrating the cell, resulting in higher levels of antibacterial resistance [36,37].

The proposed biosynthesis of the three indole alkaloids isolated in this study is based upon the bacterium’s ability to utilize indole (Figure 2 and Figure 3). *P. aeruginosa* does not typically break down tryptophan into indole via a cognate tryptophanase enzyme as other members of the Proteobacteria such as *E. coli* [38]. It is most likely that the free indole is obtained from tryptophan synthesis prior to the final condensation step between serine and indole [39]. In bacteria, indole alkaloids are mostly derived from tryptophan or its direct precursor indole, which itself is formed from chorismate through anthranilate and indole-3-glycerol-phosphate in microorganisms. Since the final step of tryptophan biosynthesis is reversible, free indole can also be formed in this catabolic process [40].

The formation of compounds **1** and **2** relies on the presence of the 4HPPD enzyme that converts 4-hydroxyphenylpyruvate into homogentisate (HGA) which can accumulate in the media resulting in the spontaneous oxidation and polymerization of HGA into HGA–melanin [41]. It was proposed that HGA–melanin acts as a catalyst for triaryl indole alkaloid production [12]. While HGA–melanin acts as the catalyst, the formation of both compounds **1** and **2** heavily relies on the condensation reaction between the C-3 of the two indole moieties and the carbaldehyde of the third moiety to create the final compound. For the biosynthesis of compound **1**, the third moiety is indole-3-carboxyaldehyde, an intermediate of indole-3-acetic acid formation, whereas for compound **2**, the third moiety is benzaldehyde, which most likely a results from aromatic hydrocarbon degradation.

The production of compound **3** trypanthrin is also heavily reliant on indole production by the bacterium, but anthranilic acid is formed as part of the *Pseudomonas* quinolone signaling (PQS) pathway [42] and isatin is formed from the combined effect of naphthalene dioxygenase (NDO) and inosine 5-monophosphate dehydrogenase (IMPD) [19]. Tryptanthrin is synthesized after the formation of indole, followed by the enzymatic oxygenation of indole into either 3-oxyindole by a monooxygenase enzyme or directly into isatin via the action of naphthalene dioxygenase. 3-oxyindole can be then further oxygenated by the action of the IMPD enzyme to also form isatin. Anthranilic acid, which comes from the PQS synthesis pathway, then forms bonds with isatin at positions N5-C5a and N11-C12 to form the quinazoline ring system.

## 4. Materials and Methods

### 4.1. General Analytical Chemistry Methods

Thin layer chromatography (TLC) of samples was performed using pre-coated 250 µm thick silica gel 60 PF254+366 aluminum-backed plates, supplied by Sigma-Aldrich (St. Louis, MI, USA). Preparative Thin Layer Chromatography (PTLC) was done using 20 cm × 20 cm glass plates coated with 250 µm- thick silica gel 60 GF254, supplied by Analtech (Newark, DE, USA). Gravity column chromatography (CC) was carried out using silica gel 60, 70–230 mesh (Sigma–Aldrich). Gel filtration (size exclusion chromatography) was performed using Lipophilic Sephadex (Sephadex^®^ LH-20) with a bead size of 25–100 µm (Sigma-Aldrich).

The compounds were visualized by exposing the TLC plates to shortwave (254 nm) and longwave (375 nm) ultraviolet light coupled with staining with PMA stain (phosphomolybdic acid (12 g) in ethanol (250 mL)).

The melting points of the isolated compounds were acquired on a Reichert micro melting point apparatus. ^1^H, ^13^C, ^1^H-^13^C HSQC, ^1^H-^13^C HMBC, ^1^H-^1^H COSY, and DEPT-135 NMR experiments were done on a Bruker 600 MHz Avance III Ultrashield Plus Spectrometer at 25 °C. Samples submitted for NMR analysis were dissolved in chloroform-d (CDCl_3_) (Sigma Aldrich). All chemical shifts were referenced to the internal standard tetramethylsilane (TMS). Mass spectroscopy was performed by Electrospray Ionization (ESI) in a Bruker MicrOTOF-Q Spectrometer (Billerica, MA, USA).

### 4.2. Bacterial Isolation and Metabolite Extraction

*Pseudomonas aeruginosa* UWI-1 was obtained from the culture collection of the Department of Life Sciences, UWI, St. Augustine. The strain was isolated during an antimicrobial bioprospecting screen and demonstrated potent antagonistic activity against several human pathogens. *P. aeruginosa* UWI-1 cultures were maintained on Kings B medium (Oxoid Ltd., Basingstoke, UK) and stored in 30% glycerol/Luria broth at −80 °C until needed.

In this study, *P. aeruginosa* UWI-1 was cultured in a shaking incubator (150 rpm) at 45 °C for 4 days in modified Luria broth (mLB) based on optimization studies (data not published). The mLB contained, per L, Beef Extract, 12.21 g; NaCl, 12.21 g; Tryptone, 5 g; Indole, 1 g; pH 6.5. mLB was inoculated with 10% seed stock in batches of 5 L for a total volume of 150 L. Extraction of the *P. aeruginosa* UWI-1 metabolites was performed using toluene in a 2:1 solvent/cultured broth ratio and repeated twice. The resulting solution was concentrated by evaporation under reduced pressure to yield 14.98 g of crude extract.

### 4.3. Bioassay-Guided Fractionation

The antibacterial activities of the extract and fractions during bioassay-guided fractionation were determined by the spot on lawn method. Methicillin-resistant *Staphylococcus aureus* (MRSA) ATCC 4827, *Escherichia coli* ATCC 25922, *Bacillus cereus* ATCC 49063, *Streptococcus pyogenes* ATCC 21,547, and *Salmonella typhimurium* serotype *enteriditis* ATCC 13,076 were used in bioassay guided fractionation. All indicator strains were maintained on Brain heart infusion (BHI) medium (Oxoid Ltd., Basingstoke, UK) and cultured at 37 °C for 18 h prior to screening.

Bacterial cell concentration was adjusted to −1.5 × 10^8^ CFU/mL in phosphate buffer (0.5 MacFarland standard) and 50 µL spread over cation-adjusted Mueller Hinton (ca-MH) agar (Oxoid Ltd., Basingstoke, UK) plates. Assays involving *S. pyogenes* were performed using ca-MH supplemented with 5% sheep’s blood. Fractions (10 µL) were dropped onto the pathogen infused plates. Plates were incubated at 37 °C for 18 h after which positive bioactivity was determined by observation of a zone of clearance.

To obtain purified active compounds, the crude toluene extract (14.98 g) was subjected to vacuum column chromatography, eluting with a gradient solvent system of petroleum ether (PE) and ethyl acetate (EtOAc) varying from 1 to 100% EtOAc. The resulting fractions were subsequently analyzed by thin layer chromatography (TLC) on silica gel plates in an 8:2 ratio of PE/EtOAc mobile phase and combined based on similarities of TLC profiles to give fourteen combined fractions (C1–14). These fractions were then screened for antibiotic activity from which four displayed inhibition against at least one of the pathogens screened.

Fraction C4 was subjected to silica gel gravity CC and eluted with PE/EtOAc (95:5) to yield **1** (4.3 g). Fractions C5 and C6 were combined and silica gel gravity CC performed eluting with PE/EtOAc (8:2) and further purified using preparative TLC using PE/EtOAc (6:4) as the mobile phase to yield **2** (1.88 g). Fraction C12 was eluted with CHCl_3_/MeOH (9:1) and further purified using Sephadex LH-20 (CHCl_3_) to yield **3** (1.092 g).

### 4.4. Assessment of Antimicrobial Activity

Antimicrobial activity of isolated compounds was determined using the minimum inhibitory concentration (MIC) following a broth microdilution assay in 96-well microplates according to the Clinical and Laboratory Standards Institute (CLSI 2014), with modifications. Bacterial cells (−10^8^ CFU mL^−1^) were inoculated into ca-MH broth containing the active compound in a twofold serial dilution series ranging between 1024 and 1 µg mL^−1^. Plates were incubated at 35 °C for 18–24 h, after which 20 µL of 5% resazurin solution was added to each well. Resazurin is a blue dye, which is reduced to a pink resofurin product by living cells. The plates were shaken for a further hour and MIC was interpreted as the well that contained the lowest concentration of test compound in which no color change was observed. The minimum bactericidal concentration (MBC) was determined by seeding 10 µL suspensions from MIC wells with no change in resazurin color onto MH agar plates and incubating for 18 h at 35 °C. The MBC was taken as the lowest concentration of the compound in which no growth was observed.

### 4.5. Biosynthetic Pathway Analysis

The Wizard Genomic DNA Purification Kit (Promega) was used to extract genomic DNA from *Pseudomonas aeruginosa* UWI-1 cells. Whole genome sequencing using the Illumina NextSeq 500 platform with paired-end read chemistry was carried out by the J. Craig Venter Institute (La Jolla, CA, USA). Contigs were mapped to the *P. aeruginosa* PAO1 reference genome (NC_002516) using Ragout v2.1.1. Annotation was done using the Prokka v1.12. The annotated genome was used to identify the presence of genes that may have contributed to the formation of the antibacterial compounds isolated in this study. The activity of these genes was compared to literature for their role in compound formation and hence putative pathways for these compounds were postulated. The whole genome shotgun project was deposited in the European Nucleotide Archive (ENA) under the project number PRJEB32405.

## 5. Conclusions

In summary, this study demonstrates the secondary metabolic versatility of *Pseudomonas aeruginosa*, as three new indole alkaloid compounds with antibacterial activity was isolated. Indole alkaloids represent a pharmaceutically interesting group of compounds which possesses a variety of biological activities. These compounds were isolated through antibacterial guided isolation and were identified as tris(1*H*-indol-3-yl)methylium (turbomycin A), bis(indol-3-yl)phenylmethane, and indolo(2, 1b) quinazoline-6, 12 dione (tryptanthrin). Antibacterial assessment of these compounds show good inhibition against Gram-positive bacteria with bactericidal activity. Most importantly, we demonstrated that *P. aeruginosa* UWI-1 possesses the genetic and metabolic means to produce these and other indole alkaloids. The biosynthesis of turbomycin A and bis(indol-3-yl)phenylmethane was formed through the condensation of two indole molecules and indole-3-carboxaldehyde for the former and benzaldehyde for the latter, catalyzed by HGA–melanin. Tryptanthrin was formed via the cyclization of anthranilic acid and isatin.

## Figures and Tables

**Figure 1 molecules-25-03744-f001:**
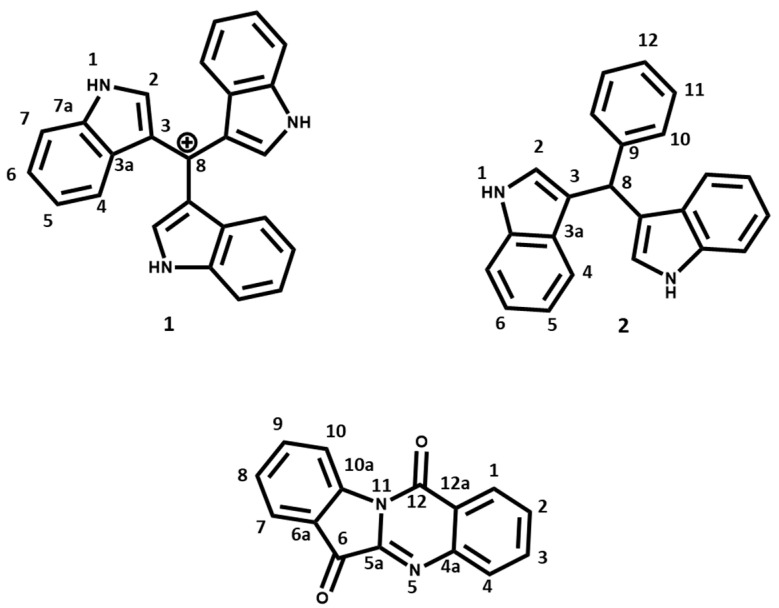
Antibacterial compounds isolated from *Pseudomonas aeruginosa* UWI-1. (**1**) tris(1*H*-indol-3-yl) methylium; (**2**) bis(indol-3-yl)phenylmethane; and (**3**) indolo (2, 1b) quinazoline-6, 12 dione (Tryptanthrin).

**Figure 2 molecules-25-03744-f002:**
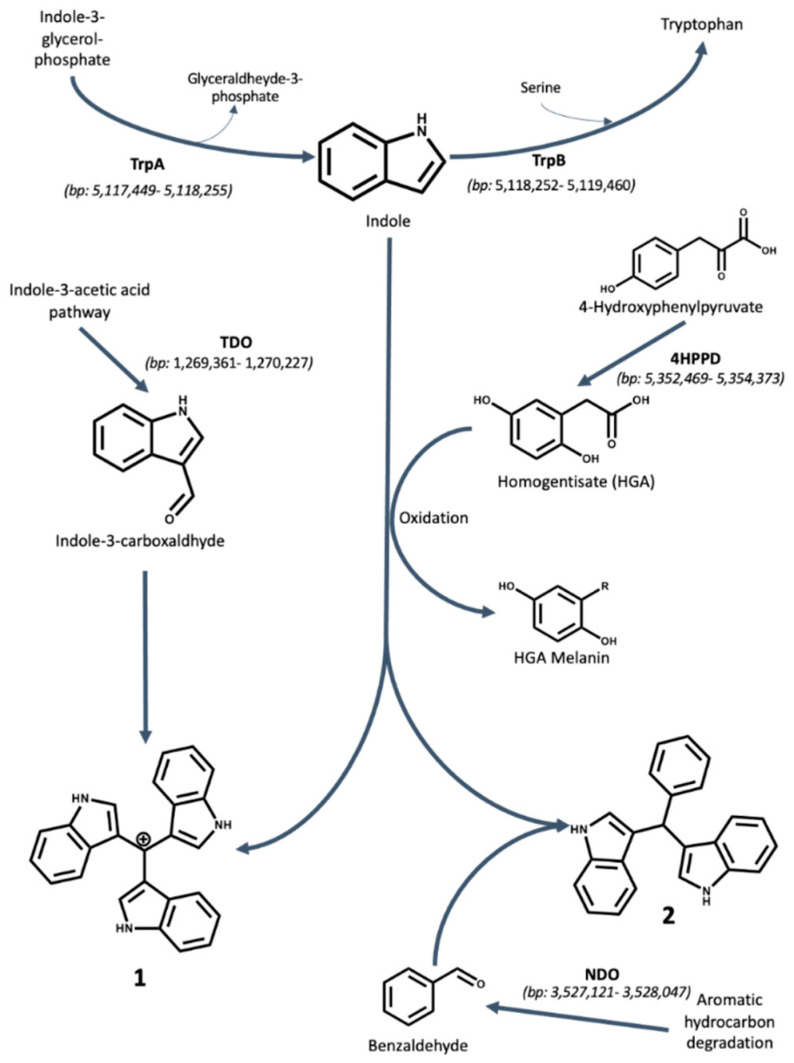
Proposed biosynthesis mechanisms for Compounds **1** and **2** by *P. aeruginosa* UWI-1. Indole is proposed to originate from tryptophan degradation via an unclassified carbon–carbon lyase. The oxidative polymerization of homogentisate (HGA) to HGA melanin serves as the catalyst for the formation of compounds **1** and **2**. Compound **1** is putatively formed via condensation of two molecules of indole and one molecule of indole-3-carboxaldehyde. Indole-3-carboxaldehyde is an intermediate product in the IAA synthesis pathway found in Pseudomonads. Compound **2** is putatively formed by the condensation of two molecules of indole and one molecule of benzaldehyde. Benzaldehyde is proposed to be derived as a product of polycyclic aromatic hydrocarbon metabolism by naphthalene 1,2-dioxygenase. The location of each gene involved in the biosynthesis is given in parentheses and is based on the genome of *P. aeruginosa* UWI-1 available from the European Nucleotide Archive (ENA) project number PRJEB32405 (https://www.ebi.ac.uk/ena/data/view/PRJEB32405).

**Figure 3 molecules-25-03744-f003:**
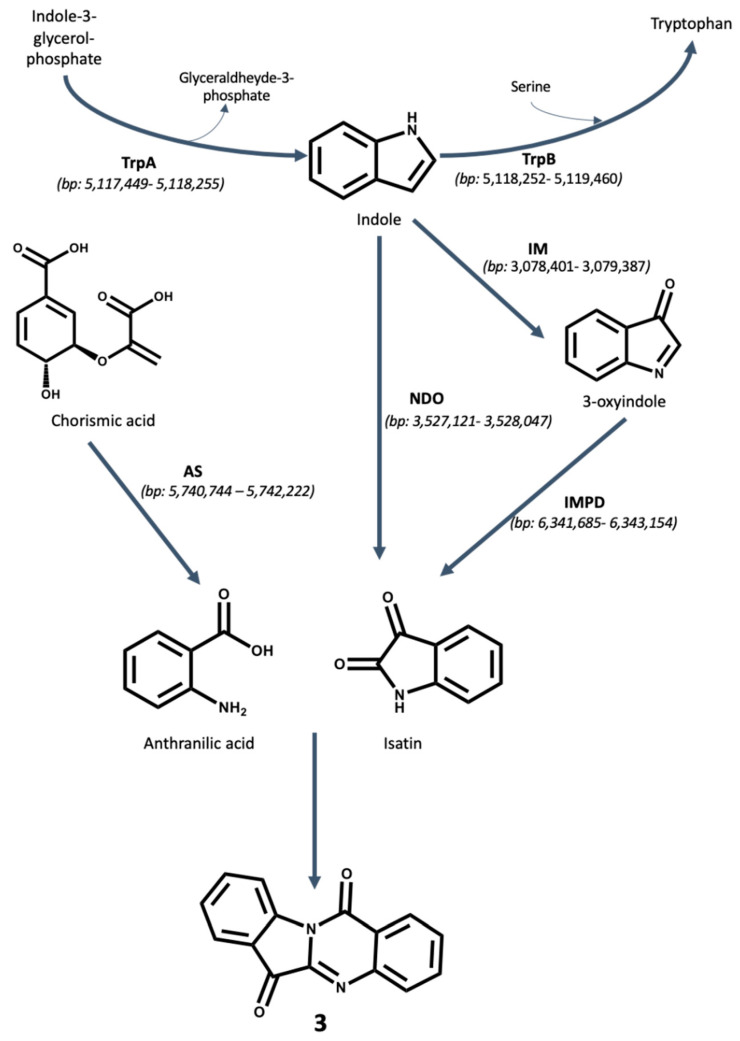
Proposed biosynthesis mechanism for tryptanthrin by *P. aeruginosa* UWI-1. Indole is proposed to originate from tryptophan degradation via an unclassified carbon–carbon lyase. Indole is oxygenated either into 3-oxyindole by indole monooxygenases (IM) and then further oxygenated by inosine 5-monophosphate dehydrogenase (IMPD) into isatin. In another route, indole may be converted directly to isatin via naphthalene 1,2-dioxygenase (NDO). Anthranilic acid is obtained through conversion of chorismic acid by the action of anthranilate synthase (AS). Both isatin and anthranilic acid fuse to form tryptanthrin. The location of each gene involved in the biosynthesis is given in parentheses and is based on the genome of *P. aeruginosa* UWI- 1 available from the European Nucleotide Archive (ENA) project number PRJEB32405 (https://www.ebi.ac.uk/ena/data/view/PRJEB32405).

**Table 1 molecules-25-03744-t001:** H and ^13^C NMR data for compounds **1** and **2**.

	Tris(1*H*-indol-3-yl)methylium (1)		Bis(indol-3-yl)phenylmethane (2)	
Position	δC	δH, (*J* in Hz)	δC	δH, (*J* in Hz)
**1**	-	8.10 (3H, brs, NH)	-	7.92 (2H, brs, NH)
**2**	102.6	7.20 (3H, s)	123.6	6.67(s, 2H)
**3**	124.1	-	136.7	-
**3a**	127.9	-	119.7	-
**4**	111.0	7.39 (3H, *J* = 8 Hz, d)	119.9	7.35 (2H, *J* = 7.5 Hz, d)
**5**	122.0	7.19 (3H, *J* = 8 Hz, t)	121.9	7.14 (2H, *J* = 7.5 Hz, t)
**6**	119.8	7.11 (3H, *J* = 8 Hz, t)	119.2	7.00 (2H, *J* = 7.5 Hz, t)
**7**	120.7	7.65 (3H, *J* = 8 Hz, d)	111.0	7.40 (2H, *J* = 7.5 Hz, d)
**7a**	135.6	-	127.0	-
**8**	109.5	-	40.2	5.89 (s, 1H)
**9**			144.0	-
**10**			128.7	7.75 (2H, *J* = 7.5 Hz, d)
**11**			128.2	7.24 (2H, *J* = 7.5 Hz, d)
**12**			126.1	7.13 (H, *J* = 7.5 Hz, d)

The chemical shifts are in δ values (ppm) from tetramethylsilane (TMS). Assignments were based on 2D NMR including HSQC and HMBC. Coupling constants were measured using ^1^H NMR in combination with phase sensitive COSY correlations. Well-resolved couplings coupling constants in Hertz (Hz) in parentheses. Chemical shifts and coupling constants were determined in CDCl_3_.

**Table 2 molecules-25-03744-t002:** H and ^13^C NMR data for compound **3** in this study.

*Tryptanthrin* (3)		
Position	δC	δH, (*J* in Hz)
**1**	127.6	8.42 (1H, *J* = 7.9 Hz, d)
**2**	130.3	7.66 (1H, *J* = 7.9 Hz, t)
**3**	135.2	7.84 (1H, *J* = 7.9 Hz, t)
**4**	130.8	8.02 (1H, *J* = 7.9 Hz, d)
**4a**	146.7	-
**5**	-	-
**5a**	144.4	-
**6**	182.5	-
**6a**	122.0	-
**7**	125.4	7.90 (1H, *J* = 8.1 Hz, d)
**8**	127.2	7.42 (1H, *J* = 8.1 Hz, t)
**9**	138.3	7.78 (1H, *J* = 8.1 Hz, t)
**10**	118.0	8.62 (1H, *J* = 8.1 Hz, d)
**11**	146.4	-
**11a**	-	-
**12**	158.1	-
**12a**	123.8	-

The chemical shifts are in δ values (ppm) from TMS. Assignments were based on 2D NMR including HSQC and HMBC. Coupling constants were measured using ^1^H NMR in combination with phase sensitive COSY correlations. Well-resolved couplings coupling constants in Hertz (Hz) in parentheses. Chemical shifts and coupling constants were determined in CDCl_3_.

**Table 3 molecules-25-03744-t003:** Antibacterial activity of compounds isolated from *Pseudomonas aeruginosa* UWI-1.

	Minimum Inhibitory Concentration/Minimum Cidal Concentration (µg mL^−1^)									
	Gram Positive					Gram Negative				
	Rods			Cocci		Rods			Cocci	
**Compound**	*Bacillus Cereus*	*Listeria Monocytogenes*	*Corynebacterium Diphtheria*	MR-*Staphylococcus Aureus*	*Streptococcus Pyogenes*	*Escherichia Coli*	*Salmonella Enteritidis*	*Klebsiella Oxytoca*	*Neisseria Meningitides*	*Haemophilus Influenza*
**1**	4/4	8/8	16/32	2/4	1/2	32/128	64/128	128/>512	32/32	32/32
**2**	64/128	128/256	32/64	32/32	64/64	NI	NI	NI	NI	NI
**3**	2/16	4/16	16/64	2/4	1/16	32/128	2/4	16/32	4/16	1/2
**Kanamycin**	32/32	64/64	128/NI	128/128	256/NI	4/4	8/16	4/16	16/16	32/32
**Erythromycin**	<1/2	4/4	NI	2/2	32/64	256/256	NI	NI	NI	128/128

NI; No inhibition observed at the concentrations tested (512–1 µg mL^−1^). Antibacterial activity was determined using the broth microdilution assay to determine the minimum inhibitory concentration (MIC) of the test compounds. Minimum bactericidal concentration (MBC) was assessed by plating samples from wells with negative growth in the MIC determination assay. Values represent the average MIC/MBC of three replicates (*n* = 3). In each case no deviation was observed and therefore the SD = ± 0.0. Kanamycin was used as a Gram-negative positive control and Erythromycin was used as a Gram-positive positive control in this study.

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
