# Peer review of "Isolation and Antibacterial Activity of Indole Alkaloids from Pseudomonas aeruginosa UWI-1"

_molecules, 2020, doi:10.3390/molecules25163744_

Round 1

Reviewer 1 Report

The authors find three compounds from Pseudomonas strain for the first time. The compounds are all known but showed antibacterial activities. The authors has proposed the biosynthetic pathway of these compouds. 

Generally, the presentation is in good organization. But the novelty is lacking.

Author Response

Authors response to reviewer 1

Thank you for the positive feedback with regards to our manuscript. Please see the following points as a response to each point raised.

  • With regards to the English language and style we have performed an extensive review of the document and have corrected any minor errors that we missed at the first submission.
  • In addressing the novelty, we acknowledge that the compounds described are not new to the literature. This is the first study that shows their association with a Pseudomonad¸ one of the most widely occurring groups of environmental bacteria globally. As such we deem that this discovery adds to the synthetic capacity of the pseudomonads. The fact too that these compounds possess antibacterial activity further emphasises this bacterium’s enhanced ecological fitness.

Reviewer 2 Report

The manuscript entitled “Isolation and antibacterial activity of indole alkaloids from Pseudomonas aeruginosa UWI-1” is written in a comprehensive way and is easy to follow, although it needs an English proofreading.

There are following issues that should be fixed:

line 39: Since MIC values depend on growth medium concentrations, you should quote the article Molecules 202025, 2947.

line 133: Exchange “almost” with “most”.

line 137: In Table 3 captions UD (undetermined) is listed, whereas it cannot be found in the Table. Please, recheck.   

Table 3 should also report standard deviations of MIC and MBC values.

Lines 212-215: Sentence is not clear, rewrite it.

Author Response

Reviewer 2 - comments and suggestions for authors

General comments:

The manuscript entitled “Isolation and antibacterial activity of indole alkaloids from Pseudomonas aeruginosa UWI-1” is written in a comprehensive way and is easy to follow, although it needs an English proofreading.

Authors response to reviewer 2:

Thank you for the positive feedback with regards to our manuscript. Please see the following points as a response to each point raised. With regards to the English language and style we have performed an extensive review of the document and have corrected any minor errors that we missed at the first submission. Some general writing improvements have been made throughout the manuscript.

Specific comments:

  1. line 39: Since MIC values depend on growth medium concentrations, you should quote the articleMolecules 2020, 25, 2947.

Response 1: We considered the relevance of this article and included it in the discussion.

  1. line 133: Exchange “almost” with “most”.

Response 2: ‘Almost’ changed to ‘most’

  1. line 137: In Table 3 captions UD (undetermined) is listed, whereas it cannot be found in the Table. Please, recheck.   

Response 3: The undetermined caption has been removed. All compounds were assessed against all bacteria.

  1. Table 3 should also report standard deviations of MIC and MBC values.

Response 4: The MIC/MBC values represented in Table 3 are the average of three replicates. Based in the MIC results however, we saw no variation in the MIC values as all the replicates gave the same results. Therefore, the standard deviation for all the MIC/MBC values are plus/minus 0, which is now indicated in the footer of the table.

  1. Lines 212-215: Sentence is not clear, rewrite it.

Response 5: Comment acknowledged, and the sentence has been re-worded.

Reviewer 3 Report

The authors have identified 3 new indoles from secondary metabolite profiles of the Pseudomonas a. They have also determined the MIC for each of these compounds. They have also carried out genomic sequencing to identify genes that can play a role in biosynthesis that leads to generation of these indoles. Overall it is a very well written and easy to understand article.

Below are some points that need to be addressed:

1. Line 73- 74: “ Also through genomic…” It is unclear what the authors are referring to here.

2. Line 132: “…comparable to the antibiotic controls, Kanamycin and Erythromycin.”  This is true for Kanamycin but not for erythromycin for the Gram Positive bacteria with the exception of corynebacterium diphtheria and Streptococus pyogenes. For some of the other gram positive bacteria Etrythromycin was similar or more potent. On the other hand Compound 1 was more potent than Erythromycin for Gram negative bacteria, but less so compared to Kanamycin. This observations need to be clearly stated.

3. Also for Table 3: How many times was this experimental set up repeated. Are these results from only n=1 experiment? If more than one experiment was performed, how do the data compare between the experimental repeats?

4. Figures showing the proposed biosynthesis mechanism should show which genes are present in the genome of pseudomonas. This will make it easier to follow the pathway.

Minor comment:

Line 66, 181: the reference seems to have an incorrect closing bracket.

Line 67 -69: reference to the previous report needs to be provided.

Line 144: concentration of 2, best to state compound 2.

Author Response

Authors response to reviewer 3:

Thank you for the positive feedback with regards to our manuscript. Please see the following points as a response to each point raised.With regards to the English language and style we have performed an extensive review of the document and have corrected any minor errors that we missed at the first submission.

Specific comments:

  1. Line 73- 74: “ Also through genomic…” It is unclear what the authors are referring to here.

Response 1: The sentence was reworded from ‘genomic investigations’ to clearly state that ‘gene sequencing and annotation coupled with gene functions comparisons with other literature’ is what was used to postulate the biosynthetic pathways.

  1. Line 132: “…comparable to the antibiotic controls, Kanamycin and Erythromycin.”  This is true for Kanamycin but not for erythromycin for the Gram Positive bacteria with the exception of corynebacterium diphtheria and Streptococus pyogenes. For some of the other gram positive bacteria Etrythromycin was similar or more potent. On the other hand Compound 1 was more potent than Erythromycin for Gram negative bacteria, but less so compared to Kanamycin. This observations need to be clearly stated.

Response 2: Line 132 has been rewritten to more clearly describe the results of Table 3. The sentence has been changed to ‘The compound was able to inhibit the growth of Gram-positive bacteria at concentrations (MIC’s = 1 - 16 µg mL-1) comparable to the antibiotic control, Kanamycin. It also performed better against C. diphtheria and S. pyogenes than Erythromycin. Compound 1 inhibited the Gram-negative bacteria as well, but only at higher concentrations (MIC’s= 32 - 128 µg mL-1). When compared to the antibiotic controls, it was observed to be more potent when compared to Erythromycin, but less so compared to Kanamycin.’

  1. Also for Table 3: How many times was this experimental set up repeated. Are these results from only n=1 experiment? If more than one experiment was performed, how do the data compare between the experimental repeats?

Response 3: The MIC/MBC values represented in Table 3 are the average of three replicates. Based in the MIC results however, we saw no variation in the MIC values therefore standard deviation was plus/minus 0/0, which is now indicated this in the footer of the table.

  1. Figures showing the proposed biosynthesis mechanism should show which genes are present in the genome of pseudomonas. This will make it easier to follow the pathway.

Response 4: The figures have been redrawn to include the genomic location (in bp) of each gene involved in the biosynthesis of the compounds.

Minor comments:

  • Line 66, 181: the reference seems to have an incorrect closing bracket.
  • Line 67 -69: reference to the previous report needs to be provided.
  • Line 144: concentration of2, best to state compound 2.

Response: All minor comments were fixed. The manuscript data for Line 67-69 is still under review and as such we have changed the sentence to state ‘unpublished report’. This data is also in a PhD thesis (under embargo) of the 1st author which can alternatively be referenced if this permissible by the journal).

Round 2

Reviewer 1 Report

Since the only new of this manuscript is that these compounds can be synthesized by the unreported microorganism, I suggest this finding can be submitted to some journal related to microbiology or chem-ecology.

Author Response

Dear Reviewer,

We the authors would like to thank you for your kind review of our manuscript. We have taken your comments into consideration and made edits based on the editors comments.

Kind Regards

Antonio

Reviewer 2 Report

The authors successfully resolved all issues raised by this reviewer. Consequently, the manuscript has been significantly improved and can be in its current version recommended for publication in Molecules.

Author Response

Dear Reviewer,

We the authors thank you for your recommendation to publish our manuscript.

Best Regards

Antonio.